# Incidence and Risk Factors for Low Birthweight and Preterm Birth in Post-Conflict Northern Uganda: A Community-Based Cohort Study

**DOI:** 10.3390/ijerph191912072

**Published:** 2022-09-23

**Authors:** Beatrice Odongkara, Victoria Nankabirwa, Grace Ndeezi, Vincentina Achora, Anna Agnes Arach, Agnes Napyo, Milton Musaba, David Mukunya, James K. Tumwine, Tylleskar Thorkild

**Affiliations:** 1Department of Paediatrics and Child Health, Faculty of Medicine, Gulu University, Gulu P.O. Box 166, Uganda; 2Centre for International Health, University of Bergen, 5020 Bergen, Norway; 3Department of Paediatrics and Child Health, School of Medicine, College of Health Sciences, Makerere University, Kampala P.O. Box 7062, Uganda; 4School of Public Health, College of Health Sciences, Makerere University, Kampala P.O. Box 7062, Uganda; 5Department of Obstetrics and Gynaecology, Faculty of Medicine, Gulu University, Gulu P.O. Box 166, Uganda; 6Department of Midwifery, Lira University, Lira P.O. Box 1035, Uganda; 7Department of Public Health, College of Health Sciences, Busitema University, Mbale P.O. Box 1460, Uganda

**Keywords:** preterm birth, low birthweight, risk factors, community-based, cohort study

## Abstract

Background: Annually, an estimated 20 million (13%) low-birthweight (LBW) and 15 million (11.1%) preterm infants are born worldwide. A paucity of data and reliance on hospital-based studies from low-income countries make it difficult to quantify the true burden of LBW and PB, the leading cause of neonatal and under-five mortality. We aimed to determine the incidence and risk factors for LBW and preterm birth in Lira district of Northern Uganda. Methods: This was a community-based cohort study, nested within a cluster-randomized trial, designed to study the effect of a combined intervention on facility-based births. In total, 1877 pregnant women were recruited into the trial and followed from 28 weeks of gestation until birth. Infants of 1556 of these women had their birthweight recorded and 1279 infants were assessed for preterm birth using a maturity rating, the New Ballard Scoring system. Low birthweight was defined as birthweight <2.5kg and preterm birth was defined as birth before 37 completed weeks of gestation. The risk factors for low birthweight and preterm birth were analysed using a multivariable generalized estimation equation for the Poisson family. Results: The incidence of LBW was 121/1556 or 7.3% (95% Confidence interval (CI): 5.4–9.6%). The incidence of preterm births was 53/1279 or 5.0% (95% CI: 3.2–7.7%). Risk factors for LBW were maternal age ≥35 years (adjusted Risk Ratio or aRR: 1.9, 95% CI: 1.1–3.4), history of a small newborn (aRR: 2.1, 95% CI: 1.2–3.7), and maternal malaria in pregnancy (aRR: 1.7, 95% CI: 1.01–2.9). Intermittent preventive treatment (IPT) for malaria, on the other hand, was associated with a reduced risk of LBW (aRR: 0.6, 95% CI: 0.4–0.8). Risk factors for preterm birth were maternal HIV infection (aRR: 2.8, 95% CI: 1.1–7.3), while maternal education for ≥7 years was associated with a reduced risk of preterm birth (aRR: 0.2, 95% CI: 0.1–0.98) in post-conflict northern Uganda. Conclusions: About 7.3% LBW and 5.0% PB infants were born in the community of post-conflict northern Uganda. Maternal malaria in pregnancy, history of small newborn and age ≥35 years increased the likelihood of LBW while IPT reduced it. Maternal HIV infection was associated with an increased risk of PB compared to HIV negative status. Maternal formal education of ≥7 years was associated with a reduced risk of PB compared to those with 0–6 years. Interventions to prevent LBW and PBs should include girl child education, and promote antenatal screening, prevention and treatment of malaria and HIV infections.

## 1. Background

Of the 140 million infants born worldwide in 2014, an estimated 20 million (13%) were born with low birthweight (<2.5 kg) [1]. Ninety percent (18/20 million) of LBW infants were born in low- and middle-income countries (LMICs) [2]. In sub-Saharan Africa, LBW prevalence varied from 7.0% to 18.0%, with the highest prevalence observed in malaria-based studies in Tanzania [3]. According to the Uganda Bureau of Statistics (UBOS) 2011, 10.4% of all live-born infants nationwide and 11.4% in the northern part of the country are LBW [4].

In 2010, an estimated 15 (uncertainty range 12–18) million preterm infants were born worldwide [5]. The global PB estimates ranges from 5% in Europe to 18% in some sub-Saharan African countries [5]. Sub-Saharan Africa and South Asia contribute 52%–60% of the global PB burden [5]. In Uganda, reports of the proportion of PBs range from 4.1% to 15% [5,6], In communities of post-conflict northern Uganda, however, its true burden is unknown.

Multiple maternal and foetal causes of LBW and/or PB (small birth size) have been described [7]. The age of the mother, either young (teenage 12–16 years) or old (≥35 years) has been linked to increased risk of small birth size [8,9]. Low maternal socio-economic and education status has been associated with small birth size [10,11,12]. Furthermore, maternal ill-health during pregnancy such as malaria and HIV infection, low body mass index (BMI) or low gestational weight gain, and hypertension have also been associated with small birth size [13,14]. A history of having given birth previously to a small infant has also been associated with LBW and/or PB recurrence in subsequent pregnancies [15,16,17]. Whereas some studies report increased risk of small birth size among women who do excessive physical work, a 2013 meta-analysis found little to no effect of the same on small birth sizes [18]. Foetal factors associated with LBW and PB include: congenital malformations, multiple foetuses, sex, and genetic factors [19,20].

In high-income countries, common causes of small birth size include provider-initiated caesarean section and assisted reproduction, [7] while in low-resource settings, it is related to maternal infections, low socio-economic status, malnutrition, and history of preterm birth or low birthweight. In post-conflict northern Uganda, however, the social disruption, lack of schooling and displacement caused by the 20 years of conflict may have modified the burden and some of the known risk factors for small birth size. Few studies exist to describe the burden of LBW and PB during the post-conflict period in northern Uganda [3].

To achieve the sustainable development goal (SDG) 3.2 target of neonatal mortality below 12 per 1000 live births by 2030, there is an urgent need to generate post-conflict context specific data on small newborns’ (LBW and PB) health burden and associated modifiable risk factors. We, therefore, aimed to (1) estimate the incidence of and (2) determine risk factors for low birthweight and preterm birth in post-conflict northern Uganda.

## 2. Methods

This was a cohort study nested within the Survival Pluss cluster randomized trial. The Survival Pluss study assessed the effect of an integrated package consisting of (i) peer support by pregnancy buddies, (ii) provision of mama (birth) kits at household level (as opposed to health facility distribution) and (iii) mobile phone messaging on facility-based births. In the trial, pregnant women were enrolled at ≥28 weeks of gestation and followed up to delivery (ClinicalTrials.gov number NCT0260505369).

The study was conducted in Lira District, Northern Uganda from July 2017 to March 2019. Lira District had a population of about 400,000 people in 2010, dwelling in 13 sub-counties, a city and 751 villages. Lira district was chosen based on its being a post-conflict area with poor maternal and child health indicators, low proportion of health facility deliveries, high neonatal mortality, and limited data on LBW and PBs burden and associated risk factors [21]. The study sites were Aromo, Agweng, and Ogur sub-counties; also chosen because they had the poorest maternal and child health indicators [9]. Each sub-county had one health centre with maternity (health centre, HC III or HC IV), and two additional lower-level health centres without maternity (HC II). Two of the HC IIIs (Agweng and Aromo), however, were not conducting deliveries before the project inception. 

A total of 1877 mothers were recruited into the trial at 28 weeks of gestation and followed up to birth. Of these, 1556 mother-infant dyads with birthweight (for LBW burden) and 1279 had both a gestational age estimate using the New Ballard Score (NBS) and birthweight (for PB burden). Only 4 persons conducted the NBS assessment, hence some infants had birthweight (from the clinic or study staff) but not gestational age estimate. 

The primary outcomes were incidence of (1) low birthweight births and (2) preterm births. Independent or exposure variables were maternal and infant factors. Maternal socio-demographic (maternal age in completed years, years of formal education, paternal occupation, marital status, wealth index groups, intervention, and domestic water source) and clinical factors (parity, HIV serostatus, malaria in pregnancy, intermittent preventive treatment (IPT) for malaria in pregnancy, history of a small newborn, multiple pregnancy, and antenatal care (ANC) attendance and infant factor (sex), were analysed for association with LBW and PB. 

A low birthweight (LBW) was defined as birthweight <2.5 kg at birth, while preterm birth (PB) was defined as being born after 28 weeks of gestation but before 37 completed weeks of gestation [22]. We calculated the incidence (risk) as the number events (LBW or PB) divided by total number of live births (population at risk), during the study period from July 2017 to March 2019, expressed as a percentage. Birthweight was measured using a digital floor scale with mother/child function (seca, Hamburg, Germany) and recorded to the nearest 2 decimal points in kilograms. Gestational age (GA) was estimated using the New Ballard Score (NBS), which employs both physical and neuromuscular maturation. The total physical maturation (PM) and neuromuscular maturation (NM), also known as maturity rating total scores (MRTS), was correlated with gestational age, recorded in completed weeks. The MRTS, ranging from −10 to 50, were then extrapolated to foetal age in weeks (20 to 44). Maternal age was recorded in completed years and categorised into three groups as 12–19, 20–34, and 35–49 years. Education was recorded in years of completed schooling and dichotomized as 0–6 and 7 or more years in school. Marital status was categorised as binary variable into ‘married’ or ‘single/separated/divorced/widowed’. Wealth index quintiles were calculated using Gini index based on several key household assets and classified ranging from the 1 ‘poorest’ to 5 ‘wealthiest’ quintiles. This was further sub-grouped into three wealth groups as follows: the lower 40% (1st–2nd quintiles), the middle 40% (3rd–4th quintiles) and the upper 20% (5th quintile). Paternal occupation was categorized during analysis as farmer, employed or unemployed. Domestic water source was categorised as ‘tap/borehole’ or ‘spring/well/river/ponds. A history of small newborn was ascertained if the answer was a ‘yes’ to the following statements: if the mother (i) mother was told by the skilled birth attendant that her infant was small at birth in the previous pregnancy based on birthweight measurement, or (ii) had history of a small infant at birth by her own assessment in prior pregnancy, or (iii) recalled the birthweight from the previous delivery which we used to categorize the infants as LBW or not, and (iv) reported that the infant was born before term in which case, we asked the mother the gestation age at birth and used it to categorise them into preterm birth (<7 months) or term (≥7 months of gestational age). Parity was the number of pregnancies the mother had before, and further re-categorised as ‘prime gravida (first time mother)’, ‘1–6′ and ‘7 or more’ children. The presence of maternal illnesses during pregnancy such as malaria or HIV were recorded as (‘yes’ ‘no’, or ‘unknown’) based on antenatal test results. Antenatal care (ANC) attendance was recorded as ‘yes’ if the woman attended antenatal clinic at least once during the current pregnancy. Maternal malaria IPT in pregnancy was recorded as ‘yes’ if the mother received intermittent preventive treatment for malaria during pregnancy. Intervention was recorded as ‘yes’ if the mother received the Survival Pluss intervention package (mama kit, SMS, and peer buddies) during pregnancy. We analysed sub-samples of mother-infant pairs from the Survival Pluss cohort who had infants with birthweight (1556) or both birthweight and gestational age by NBS assessment (1279), respectively. We compared the included to the excluded sample and there was minimal difference in baseline socio-demographic characteristics between the analysed and excluded groups except for maternal age in the PB sample and health facility delivery and father’s occupation in the LBW sample, (Table 1). The Survival Pluss study included and followed all pregnant women in the participating communities from 28 weeks of gestation, who had no intention of moving away from the study area within a year of enrolment and who had no psychiatric illness that could inhibit the informed consent process. We excluded infants whose parents declined newborn examinations, those who died at birth or who had severe congenital abnormalities (anencephaly and exomphalos) and those without birthweight (for LBW) and without birthweight and NBS (for PBs).

### 2.1. Study Procedures

Prior to recruitment, research assistants were trained on the study protocol, weight measurement, and electronic data collection tool, the open data kit (ODK) software (https://opendatakit.org/ (accessed on 6 December 2017)), and the New Ballad Scoring system (NBS) for gestational age assessment. Pregnant mothers were identified by community recruiters who informed the study team. The research assistants were then dispatched to see the identified mothers. Those who met the inclusion criteria were consented and recruited. The enrolled pregnant women were followed up to birth and postnatally to two and seven days, for birthweight and administration of the NBS, respectively. The neonatal anthropometrics (birthweight) and NBS were done within two days and seven days for accurate determination of birthweight and gestation age, respectively. After birth, the same recruiters informed the study team who in turn visited the mother-infant dyads at birth for delivery questionnaire administration and anthropometric (birthweight, length, head, chest and abdominal circumferences) measurements. The weighing scales and length/height boards were calibrated before each field visit and before each measurement was taken. The weighing scales were checked for accuracy daily with known standard weights. Data was collected using standardized pre–coded questionnaires in ODK, and immediately sent to the server for safe custody by the data manager. Data cleaning and checking for completeness were done for quality control throughout the data collection process.

A total of four research nurses and midwives were trained on the NBS tool. The overall intra-rater (percentage agreement: 82.56%, kappa: 0.806, 95% CI: 0.788–0.823) and inter-rater (percentage agreement: 77.5%; kappa: 0.774, 95% CI: 0.613–0.936) reliability for the Ballard scoring tool were strong. The principal investigator (BO) worked with and supervised the research assistants on data collection and documentation.

### 2.2. Statistical Analysis

The data collected using ODK was sent to a server from where it was downloaded to Stata 14 (Stata Corp, College Station, TX, USA) for analysis. The incidence of LBW and PB were sex standardized and cluster adjusted and presented as the proportion of LBW and PBs to the total number of live births reported in percent (see Table 2 in Results Section. Descriptive statistics for categorical variables were summarized into proportions and the results presented in (see Table 3 and Table 4, Results Section). Inferential statistics (the risk factors for LBW and PB), were analysed using bivariable and multivariable generalised estimation equation for the binary categorical outcome of LBW and PB (see Table 3 and Table 4 in Results Section). Significant factors with *p* value ≤ 0.05 at bivariable analysis were taken into the multivariable generalized estimation equation model with a log link to Poisson family, adjusting for clustering and potential confounding. Known risk factors for LBW and PB such as infant sex, wealth index, and integrated intervention were also added into the final model. The crude and adjusted risk ratios were compared during the multivariable regression analysis. A difference of ≥10% between crude and adjusted risk ratios were considered confounding.

## 3. Results

### 3.1. Study Profile

Of the 1877 pregnant women recruited into Survival Pluss trial, 44 were lost to follow-up, 277 had missing birthweight and further 277 were not reached in time for gestational age estimation by NBS. Of those with birthweight, 7.8% (121/1556) were LBW and of those with gestational age estimate, 4.1% (53/1279) were assessed to be born preterm. Of the LBW infants with gestational age, 19% (20/105) were considered preterm while 37.7% (20/53) of preterm infants were low birthweight (Figure 1).

### 3.2. Baseline and Clinical Characteristics of Study Participants

Of the 1556 mother-infant dyads, a quarter of the mothers were first time mothers (prime gravida), 22 (1.4%) were twins, and 90% were married. Most of the fathers were subsistence farmers. Most families used tap or borehole water for domestic consumption. Around 4.4% of the mothers were HIV seropositive, while up to 4.6% did not know their HIV status. Close to 16.9% of mothers had prior history of small newborn in the most recent (second last) delivery. The male to female ratio approximated 1:1, Table 1.

### 3.3. The Incidence of Low Birthweight and Preterm Birth 

#### 3.3.1. Low Birthweight 

The number of low birthweight infants was 121/1556, 7.7%. The sex and cluster adjusted incidence of LBW in post-conflict northern Uganda was 7.3% (95% Confidence interval (CI): 5.4%–9.6%).

#### 3.3.2. Preterm Birth

The incidence of preterm births assessed by NBS was 53/1279 or 4.1%. The sex and cluster adjusted incidence of PB in post-conflict northern Uganda was 5.0% (95% CI: 3.2%–7.7%). The New Ballard Score being subjective, we analysed in a sensitivity analysis, the effect of potential systematic over–scoring of the maturity rating total score (MRTS) on the incidence of preterm birth (Table 2). The crude and the sex and cluster adjusted incidence of preterm birth is presented in case the infants were over–scored by 1, 2, 3, or 4 MRTS. 

**Table 2 ijerph-19-12072-t002:** Sensitivity analysis of the incidence of preterm birth based on the New Ballard among 1279 infants in Northern Uganda.

	Crude Incidence of Preterm Birth (95% CI)	Cluster and Adjusted Incidence of Preterm Birth (95% CI)
Using the original New Ballard Score	4.1% (3.0–5.8%)	5.0% (3.2–7.7%)
Subtracting 1 score point from the New Ballard Score	5.5% (4.4–6.9%)	6.4% (4.4–9.2%)
Subtracting 2 score points from the New Ballard Score	7.8% (6.5–9.6%)	8.6% (6.1–12.2%)
Subtracting 3 score points from the New Ballard Score	12.1% (10.4–14.0%)	13.1% (10.0–16.9%)
Subtracting 4 score points from the New Ballard Score	17.1% (15.2–19.3%)	17.8% (14.6–21.4%)

CI confidence interval.

### 3.4. Risk Factors for Low Birthweight and Preterm Birth

#### 3.4.1. Low Birthweight

The factors that were associated with increased risk of a low birthweight infants in our cohort were advanced maternal age (≥35 years), history of a small newborn in prior pregnancy, malaria infection, and unknown malaria status in pregnancy (Table 3). Infants born to mothers aged 35 or more years were two (adjusted RR 1.9 (95% CI: 1.1 –3.9) times more likely to be LBW compared to those born to mothers aged 20–34 years. A history of a small newborn in the second last pregnancy doubled the risk (aRR: 2.1, 95% CI: 1.2–3.4) of LBW compared to those without. A positive malaria test (aRR: 1.7, 95% CI: 1.01–2.9) or an unknown malaria status during pregnancy (aRR 1.9, 95% CI: 1.1–3.2) almost doubled the risk of LBW among the infants compared to those with known malaria negative tests. On the other hand, infants whose mothers received intermittent preventive treatment for malaria during pregnancy had a 40% (aRR 0.6, 95% CI: 0.4–0.8) reduced risk of being LBW compared to those who did not. The integrated intervention package had no effect on the LBW in this post conflict setting of northern Uganda. These and more details are summarized in Table 3. Similarly, other known risk factors for LBW such as poverty, maternal education, teenage motherhood, grand multi–parity, ANC attendance and HIV infection were not associated with an increased risk of LBW among mothers in the cohort. 

**Table 3 ijerph-19-12072-t003:** Bi- and multi-variable analysis of risk factors for low birthweight in northern Uganda.

Characteristics	All N = 1556 n (%)	LBW N = 121 n (%)	Crude RR (95% CI) N = 1556	*p* Value	Adjusted RR (95% CI) N = 1556	*p* Value
Maternal characteristics						
Maternal age						
12–19 years	415 (26.7)	40 (33.1)	1.4 (1.0–2.0)	0.048	1.3 (0.8–2.1)	0.351
20–34 years	982 (63.1)	67 (55.4)	Ref			
≥35 years	159 (10.2)	14 (11.6)	1.3 (0.9–1.9)	0.183	1.9 (1.1–3.4)	0.021
Maternal education						
0–6 years	1246 (80.1)	91 (75.2)	Ref			
≥7 years	310 (19.9)	30 (24.8)	1.3 (0.9–2.0)	0.190	1.4 (0.9–2.3)	0.102
Maternal vocational education						
No	1371 (88.1)	103 (85.1)	Ref			
Yes	185 (11.9)	18 (14.9)	1.3 (0.8–2.1)	0.297		
Marital status						
Married	1417 (91.1)	110 (90.9)	1.0 (0.5–1.8)	0.951		
Single/separated/divorced/widowed	139 (8.9)	11 (9.1)	Ref			
Wealth index groups						
Lower 40%	708 (45.5)	62 (51.2)	Ref			
Middle 40%	547 (35.2)	40 (33.1)	0.8 (0.6–1.3)	0.379	0.8 (0.6–1.3)	0.402
Upper 20%	301 (19.3)	19 (15.7)	0.7 (0.5–1.2)	0.171	0.7 (0.4–1.2)	0.255
Father’s occupation						
Farmer	1058 (68.0)	87 (71.9)	Ref			
Employed	348 (22.4)	22 (18.2)	1.0 (0.5–1.8)	0.929		
Unemployed	150 (9.6)	12 (9.9)	0.8 (0.5–1.2)	0.237		
Domestic water source						
Tap/Borehole	977 (62.8)	72 (59.5)	Ref			
Spring/river/well/stream/pond	579 (37.2)	49 (40.5)	1.1 (0.8–1.7)	0.476		
Intervention						
No	740 (47.6)	60 (49.6)	Ref			
Yes	816 (52.4)	61 (50.4)	0.9 (0.6–1.3)	0.656	0.9 (0.6–1.4)	0.716
Facility Delivery						
No	482 (31.1)	42 (34.7)				
Yes	1070 (68.9)	79 (65.3)	0.8 (0.6–1.1)	0.251		
Maternal clinical characteristics						
History of a small infant						
No	218 (14.0)	19 (15.7)	Ref			
Yes	985 (63.3)	68 (56.2)	1.3 (0.7–2.1)	0.386	2.1 (1.2–3.7)	0.014
Prime gravida	353 (22.7)	34 (28.1)	1.4 (0.9–2.1)	0.090	1.1 (0.6–1.8)	0.778
Parity						
Prime gravida	353 (22.7)	34 (28.1)	Omitted			
1–6	1043 (67.0)	77 (63.6)	Ref			
7 or more	160 (10.3)	10 (8.3)	0.8 (0.5–1.5)	0.573	0.6 (0.3–1.4)	0.226
Maternal HIV infection						
No	1455 (93.5)	116 (95.9)	Ref			
Yes	73 (4.7)	5 (4.1)	0.9 (0.4–2.0)	0.723	0.9 (0.4–1.8)	0.719
Unknown	28 (1.8)	0 (0.0)	Not applicable			
Antennal attendance						
No	352 (22.6)	30 (24.8)	Ref			
Yes	1204 (77.4)	91 (75.2)	0.9 (0.6–1.3)	0.522		
IPT for malaria in pregnancy						
No	704 (45.2)	69 (57.0)	Ref			
Yes	852 (54.8)	52 (43.0)	0.6 (0.4–0.8)	0.003	0.6 (0.4–0.8)	0.001
Malaria in pregnancy						
No	502 (32.3)	25 (20.7)	Ref			
Yes	388 (24.9)	32 (26.4)	1.7 (1.01–2.7)	0.046	1.7 (1.01–2.9)	0.045
Unknown	666 (42.8)	64 (52.9)	1.9 (1.2–3.0)	0.005	1.9 (1.1–3.2)	0.020
Infant sex						
Female	757 (48.7)	63 (52.1)	Ref			
Male	799 (51.3)	58 (47.9)	0.9 (0.6–1.2)	0.393	0.9 (0.7–1.2)	0.463

N/n (%) frequency (percentage), RR risk ratio, CI confidence interval, HIV human immunodeficiency virus.

#### 3.4.2. Preterm Birth

HIV infection was associated with an increased risk of PB (adjusted or aRR: 2.9, 95% CI: 1.1–7.3) in the multivariable analysis (Table 4). Maternal education (≥7 years) was associated with a reduced risk of PB (aRR: 0.3, 95% CI: 0.1–0.98).

**Table 4 ijerph-19-12072-t004:** Bivariable and multivariable analysis of risk factors for preterm birth in northern Uganda.

Characteristics	All N = 1279 n (%)	PB N = 53 n (%)	Crude RR (95% CI) N = 1279	*p* Value	Adjusted RR (95% CI) N = 1279	*p* Value
Maternal characteristics						
Maternal age						
12–19 years	330 (25.8)	18 (34.0)	1.6 (0.9–2.9)	0.142	2.0 (1.0–4.3)	0.050
20–34 years	815 (63.7)	28 (52.8)	Ref			
≥35 years	134 (10.5)	7 (13.2)	1.5 (0.7–3.5)	0.295	1.2 (0.6–2.6)	0.612
Maternal education						
0–6 years	1032 (80.7)	50 (94.3)	Ref			
≥7 years	247 (19.3)	3 (5.7)	0.2 (0.1–0.8)	0.022	0.3 (0.1–0.98)	0.047
Maternal vocational education						
No	1131 (88.4)	45 (84.9)				
Yes	148 (11.6)	8 (15.1)				
Marital status						
Married	1166 (91.2)	47 (88.7)	0.7 (0.3–1.5)	0.393		
Single/separated/divorced/widowed	113 (8.8)	6 (11.3)	Ref			
Wealth index						
Lower 40%	574 (44.9)	26 (49.1)	Ref			
Middle 40%	465 (36.3)	18 (34.0)	0.8 (0.5–1.4)	0.513	0.9 (0.6–1.5)	0.815
Upper 20%	240 (18.8)	9 (17.0)	0.8 (0.4–1.9)	0.650	1.1 (0.5–2.5)	0.847
Father’s occupation						
Farmer	883 (69.0)	38 (71.7)	Ref			
Employed	274 (21.4)	8 (15.1)	1.4 (0.7–2.9)	0.342		
Unemployed	122 (9.5)	7 (13.2)	0.7 (0.4–1.4)	0.305		
Domestic water source						
Tap/Borehole	802 (62.7)	27 (50.9)	Ref			
Spring/river/well/stream/pond	477 (37.3)	26 (49.1)	1.1 (0.8–1.7)	0.476	1.5 (0.9–2.6)	0.121
Intervention						
No	601 (47.0)	23 (43.4)	Ref			
Yes	678 (53.0)	30 (56.6)	1.1 (0.6–2.1)	0.670	1.2 (0.7–2.2)	0.517
Facility Delivery						
No	397 (31.0)	23 (4.4)	Ref			
Yes	882 (69.0)	30 (56.6)	0.6 (0.3- 1.01)	0.054	0.6 (0.4–1.0)	0.045
Maternal clinical factors						
History of a small infant						
No	964 (75.4)	39 (73.6)	Ref			
Yes	40 (3.1)	2 (3.8)	1.2 (0.2–5.7)	0.927	1.0 (0.2–5.2)	0.986
Prime gravida	275 (21.5)	12 (22.6)	1.1 (0.5–2.0)	0.884	0.8 (0.3–1.8)	0.557
Parity						
Prime gravida	275 (21.5)	12 (22.6)	Ref			
1–6	872 (68.2)	34 (64.2)	1.1 (0.6–2.1)	0.790		
7 or more	132 (10.3)	7 (13.2)	1.4 (0.7–2.6)	0.346		
Maternal HIV infection						
No	1205 (94.2)	47 (88.7)	Ref			
Yes	61 (4.8)	6 (11.3)	2.2 (0.9–5.6)	0.094	2.9 (1.1–7.3)	0.026
Unknown	13 (1.0)	0 (0.0)	NA			
Antenatal attendance						
No	283 (22.1)	14 (26.4)	Ref			
Yes	996 (77.9)	39 (73.6)	0.8 (0.4–1.4)	0.451		
IPT for malaria in pregnancy						
No	695 (54.3)	29 (54.7)	Ref			
Yes	584 (45.7)	24 (45.3)	0.9 (0.5–1.6)	0.832	1.0 (0.6–1.8)	0.886
Malaria in pregnancy						
No	330 (25.8)	15 (28.3)	Ref			
Yes	342 (26.7)	13 (24.5)	0.8 (0.5–1.5)	0.568		
Unknown	607 (47.5)	25 (47.2)	0.9 (0.5–1.6)	0.785		
Infant sex						
Female	620 (48.5)	20 (37.7)	Ref			
Male	659 (51.5)	33 (62.3)	1.6 (0.9–2.7)	0.117	1.6 (1.0–2.8)	0.070

N/n (%) frequency (percentage), RR risk ratio, CI confidence interval, PB preterm birth, NA not applicable, IPT intermittent preventive treatment, HIV human immunodeficiency virus.

## 4. Discussion

In our cohort, the incidence of LBW was 7.3%. The proportion of LBW in post-conflict rural Northern Uganda is lower than most other estimates, be it the global, sub-Saharan Africa, or Uganda [1,23,24]. This study was a sub-study of a trial in which one of the inclusion criteria was a gestational age 28 or more weeks of pregnancy. Given that women were enrolled at 28 or more weeks, low birthweight occurring before recruitment were systematically excluded. Therefore, our study is likely to have underestimated the true incidence of both LBW.

Factors associated with low birthweight included maternal age ≥35 years, history of a small newborn in the previous pregnancy, maternal malaria in pregnancy and intermittent preventive treatment (IPT) for malaria. The finding that advanced maternal age (≥35 years) was associated with an increased risk of LBW in our cohort is not unique to our report. Numerous studies have described the increased risk of LBW with low or advanced maternal age [25,26]. The study also reports an associated increased risk of LBW among mothers with history of a small newborn, in the most recent pregnancy. Other studies report similar links [17,27].

The relationship between malaria in pregnancy and its association with increased risk of LBW has been reported elsewhere [28]. Similarly, we also report reduced risk of LBW among infants born to mothers who had intermittent preventive therapy for malaria during pregnancy. Malaria IPT during pregnancy reduces placental malaria, a long known risk factor for LBW and preterm births (small newborn) [29].

The preterm birth (PB) proportion in our cohort was 5.0% and is similar to a hospital-based study in Eastern Uganda, with similar inclusion and exclusion criteria [6]. The observed estimate in this cohort, however, is lower than the global, sub-Saharan Africa, or Uganda estimates [5,24].

The low PB proportion observed in our study may be due to the trial eligibility criteria discussed above that could have resulted in exclusion of some preterm births occurring before recruitment into the main trial. Secondly, the NBS for foetal maturation for gestational age determination (instead of mid-pregnancy ultrasound as the gold standard), may have contributed to the underestimation of PB in this cohort. For instance, a study by Sasidharan and colleagues reported that NBS overestimated gestational age (GA) by up to 2 weeks (8 MRTS), with increasing postnatal age [30]. Therefore, if the current global PB modelled estimates by the global burden of disease (GBD) research group are true, we may have over-estimated GA by 3MRTS (1.2 weeks), see our sensitivity analysis in Table 2 above. Although scientists modified the NBS system to identify extremely preterm babies up to seven days of postnatal age, it seems postnatal age at assessment may have played a role in the PB estimates in our cohort. The exclusion of 363/1833 (19.8%) infants not reached for NBS gestational age (GA) assessment within 7 days of postnatal life, and another 191/1833 (10.4%) of the infants without birthweight, may have also resulted in the observed low PB incidence proportion. Despite the challenges faced in PB diagnosis in our setting, the findings may still be relevant in contributing to the pool of knowledge on preterm births and associated risk factors, to guide decision making in a resource-limited post-conflict setting.

Factors associated with an increased risk of preterm birth include maternal HIV infection. Maternal education for seven or more years was associated with a reduced risk. Our finding that low maternal education is associated with an increased risk of PB has been reported elsewhere [31,32,33]. The increased risk of PBs among HIV infected women, compared to the uninfected has also been documented over the last 3 decades [34].

In our cohort, teenage motherhood doubled the risk of PB and this is of public health importance. The finding is similar to findings from several other studies across the globe [35,36]. Although the biological link between teenage pregnancy and PB is not properly understood, [10,37] pregnant teens are likely to be disfavoured in several aspects such as education, access to care and nutrition compared to older mothers [38,39,40].

The study also reported an increased risk of PB among male infants, compared to female infants. This may be a methodological artefact due to differences in NBS scoring of the two sexes. An analysis of mean difference for the overall MRTS and individual elements for physical and neuromuscular scores by sex, demonstrated a significant difference in physical maturity rating for breasts. Female infants were systematically over-scored by 0.14 (95% CI: 0.08–0.21) equivalent to 4 days (95% CI: 2–6) points in the physical maturity rating for breasts, which may contribute to fewer infants being classified as being PB. It is still possible that there is still true increase in the risk of PB for male infants as this has been reported elsewhere [19,41].

## 5. Limitations and Strengths

The main limitation of our study is the potential for selection bias at inclusion which may have introduced systematic error. In the main Survival Pluss randomised trial in which our observational study was nested, inclusions were allowed at any time from 28 or more weeks of gestation (WoG). The inclusion of pregnancies from 28 or more WoG is based on foetal viability in our low resource settings. Deliveries before 28 weeks of gestational age are considered abortions (in-service personal experience). It means that a pregnant woman could be included at, for instance, 35 weeks of gestation. This also means that not all pregnant women in the study area were followed up from exactly 28 WoG. Women who had LBW and PB before recruitment into the trial were systematically excluded from our study. This likely caused us to underestimate the true incidence of LBW and PB. This could explain the low incidence of LBW and PB reported in this study. 

Furthermore, additional selection biases could have occurred due to loss to follow up resulting from missing birthweight and/or gestational age assessment (GA) of the infants. For the PB, we restricted the analysis to the sample of infants with both GA and birthweight. Approximately 554 infants (30%) of the 1833 in the cohort did not have both birthweight and gestational age measurements and were excluded from the analysis. This could have possibly resulted in a selection bias. That said, in a sensitivity analysis, we found no major differences in socio-demographic characteristics of included and excluded participants. Future studies to estimate the incidence of LBW and PB should aim at enrolling mothers in the first trimester and following up the entire cohort for the remainder of the pregnancy. This would permit more accurate gestational age estimations and provide a more complete cohort. 

Albeit the above limitations, there were several strengths in our study. Firstly, we used a community-based cohort—likely to reflect the community at large. Secondly, we were able to follow-up and obtain birthweight within 48 hours on 1556/1833 (85%) of the cohort, minimising the risk of selection bias. Thirdly, mothers were interviewed shortly after the delivery, minimising the likelihood of recall bias. Lastly, we used hard, explicitly defined outcome measures (low birthweight and preterm birth). This reduced the likelihood of misclassification/information bias.

## 6. Conclusions

The incidence of LBW and PB were low, compared to the national, sub-Saharan Africa and global estimates. Advanced maternal age of ≥35 years and history of a small newborn were associated with increased risk of low birthweight. Maternal formal education for ≥7 years was associated with a reduced risk of PB while HIV infection was associated with an increased risk of PB.

## Figures and Tables

**Figure 1 ijerph-19-12072-f001:**
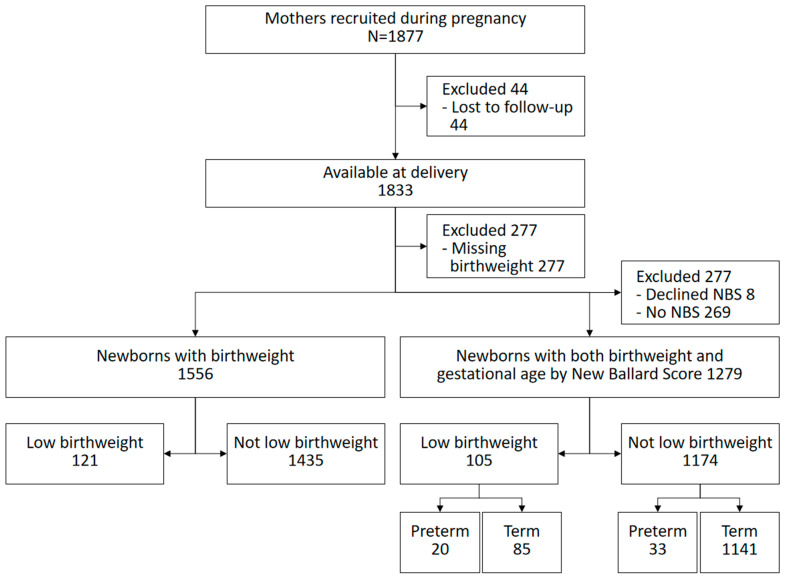
Study profile.

**Table 1 ijerph-19-12072-t001:** Comparison of baseline characteristics between included and excluded study participants in the two analyses—low birthweight and preterm birth—in Northern Uganda.

Characteristics	Low Birthweight				Preterm Birth			
	All N = 1877 n (%)	Analysed N = 1556 n (%)	Excluded N = 321 n (%)	*p* Value	All N = 1877 n (%)	Analysed N = 1279 n (%)	Excluded N = 598 n (%)	*p* Value
Maternal characteristics								
Maternal age								
12–19 years	510 (27.2)	415 (26.7)	95 (29.6)		510 (27.2)	330 (25.8)	180 (30.1)	
20–34 years	1174 (62.5)	982 (63.1)	192 (59.8)	0.325	1174 (62.5)	815 (63.7)	359 (60.0)	0.017
≥35 years	193 (10.3)	159 (10.2)	35 (10.6)		193 (10.3)	134 (10.5)	59 ( 9.9)	
Maternal education								
0–6 years	1515 (80.7)	1246 (80.1)	269 (83.8)		1515 (80.7)	1032 (80.7)	483 (80.8)	
≥7 years	362 (19.3)	310 (19.9)	52 (16.2)	0.117	362 (19.3)	247 (19.3)	115 (19.2)	0.896
Maternal vocational education								
No	1663 (88.6)	1371 (88.1)	292 (92.0)		1663 (88.6)	1131 (88.4)	532 (89.0)	
Yes	214 (11.4)	185 (11.9)	29 ( 8.9)	0.224	214 (11.4)	148 (11.6)	66 (11.0)	0.700
Marital status								
Married	1708 (91.0)	1417 (91.1)	291 (90.7)	0.495	1708 (91.0)	1166 (91.2)	542 (90.6)	0.557
Single/separated/divorced/widow	169 ( 9.0)	139 ( 8.9)	30 ( 9.3)		169 ( 9.0)	113 (8.8)	56 ( 9.4)	
Wealth index								
Lower 40%	837 (44.6)	708 (45.5)	129 (40.2)		837 (44.6)	574 (44.9)	263 (44.0)	
Middle 40%	665 (35.4)	547 (35.2)	118 (36.8)	0.329	665 (35.4)	465 (36.4)	200 (33.4)	0.139
Upper 20%	375 (20.0)	301 (19.3)	74 (23.0)		375 (20.0)	240 (18.8)	135 (22.6)	
Father’s occupation								
Farmer	1275 (67.9)	1058 (68.0)	217 (67.6)		1275 (67.9)	883 (69.1)	392 (65.5)	
Employed	390 (20.8)	348 (22.4)	42 (13.1)	0.022	390 (20.8)	274 (21.4)	116 (19.4)	0.688
Unemployed	168 ( 9.0)	150 ( 9.6)	18 ( 5.6)		168 ( 9.0)	122 ( 9.5)	46 ( 7.7)	
Missing	44 ( 2.3)	0 ( 0.0)	44 (13.7)		44 ( 2.3)	0 ( 0.0)	44 ( 7.4)	
Domestic water source								
Tap/Borehole	1188 (63.3)	977 (62.8)	211 (65.7)	0.459	1188 (63.3)	802 (62.7)	386 (64.6)	0.268
Spring/river/well/stream/pond	689 (36.7)	579 (37.2)	110 (34.3)		689 (36.7)	477 (37.3)	212 (35.4)	
Intervention								
No	855 (47.2)	740 (47.6)	145 (45.2)		885 (47.2)	601 (47.0)	284 (47.5)	
Yes	992 (52.9)	816 (52.4)	176 (54.8)	0.625	992 (52.8)	678 (53.0)	314 (52.5)	0.956
Facility Delivery								
No	644 (34.3)	484(31.1)	160 (49.8)		644 (34.3)	397 (31.0)	247 (41.3)	
Yes	1233 (65.7)	1072(68.9)	161 (50.2)	0.000	1233 (65.7)	882 (67.0)	351 (58.7)	0.000
Maternal clinical characteristics								
History of small infant								
No	1131 (60.2)	985 (63.3)	146 (45.5)		1131 (60.3)	964 (75.4)	167 (30.2)	
Yes	317 (16.9)	218 (14.0)	99 (30.8)	0.000	317 (16.9)	40 ( 3.1)	277 (50.0)	0.000
Prime gravida	429 (22.9)	353 (22.7)	76 (23.7)		429 (22.9)	275 (21.5)	154 (27.8)	
Parity								
Prime gravida	429 (22.9)	353 (22.7)	76 (23.7)		429 (22.9)	275 (21.5)	154 (25.7)	
1–6	1257 (67.0)	1043 (67.0)	214 (66.8)	0.857	1257 (67.0)	872 (68.2)	385 (64.4)	0.025
7 or more	191 (10.2)	160 (10.3)	31 ( 9.7)		191 (10.2)	132 (10.3)	59 ( 9.9)	
Maternal HIV infection								
No	1708 (91.0)	1455 (93.5)	253 (78.8)		1708 (91.0)	1205 (94.2)	503 (84.1)	
Yes	83 ( 4.4)	73 ( 4.7)	10 ( 3.1)	0.000	83 ( 4.4)	61 ( 4.8)	22 ( 6.7)	0.000
Unknown	86 ( 4.6)	28 ( 1.8)	58 (18.1)		86 ( 4.6)	13 ( 1.0)	73 (12.2)	
Antenatal attendance								
No	395 (21.0)	352 (22.6)	43 (13.4)		395 (21.0)	283 (22.1)	112 (18.7)	
Yes	1482 (79.0)	1204 (77.4)	278 (86.6)	0.000	1482 (79.0)	996 (77.9)	486 (81.3)	0.088
IPT ^a^ for malaria in pregnancy								
No	764 (40.7)	704 (45.2)	60(18.7)		764 (40.7)	695 (54.3)	69 (11.5)	
Yes	1113 (59.3)	852 (54.8)	261 (81.3)	0.000	1113 (59.3)	584 (45.7)	529 (88.5)	0.000
Maternal malaria in pregnancy								
No	602 (32.1)	502 (32.3)	100 (31.2)		602 (32.1)	272 (45.5)	330 (25.8)	
Yes	459 (24.4)	388 (24.9)	71 (22.1)	0.245	459 (24.4)	117 (19.6)	342 (26.7)	0.000
Unknown	816 (43.5)	666 (42.8)	150 (46.7)		816 (43.5)	209 (35.0)	607 (47.5)	
Infant sex								
Female	892 (47.5)	757 (48.7)	135 (42.0)		892 (47.5)	620 (48.5)	272 (45.5)	
Male	943 (50.2)	799 (51.3)	144 (44.9)	0.950	943 (50.2)	659 (51.5)	284 (47.5)	0.816
Missing	42 ( 2.3)	0 ( 0.0)	42 (13.1)		42 (2.2)	0 (0.0)	42 (7.0)	

N/n (%) frequency (percentage), ^a^ IPT = Intermittent preventive treatment for malaria.

## Data Availability

The data for this manuscript may be access from the corresponding author on reasonable request. The corresponding author’s email: beachristo2003@gmail.com and Tel.: +256772896397.

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
