# Peer review of "Incidence and Risk Factors for Low Birthweight and Preterm Birth in Post-Conflict Northern Uganda: A Community-Based Cohort Study"

_ijerph, 2022, doi:10.3390/ijerph191912072_

Round 1

Reviewer 1 Report

In background: the 5th paragraph mentions SDG. what is that? I can’t seem to find that (maybe I overlooked it).

you have many tables. Will you be putting lines with these tables because they are a little difficult to read without the lines.

Under Discussion, 5th paragraph you mention Paper I for sensitivity analysis. Where would the reader find this if your manuscript was published?

You mention this in your Limitations and strengths section: that your observational study was nested in a larger study. But are you able to clarify for the reader why you chose 28 weeks? Could you not start earlier?

Interesting manuscript. I am blessed to live in the USA, so to read that Uganda can do studies, that malaria and HIV can be prevalent, that you actually got respondents to answer was all great.

Author Response

Reviewer 1

Thank you so much for your comments. Here below is the point by point response to the comments

Comments and Suggestions for Authors

Question (Qn): In background: the 5th paragraph mentions SDG. What is that? I can’t seem to find that (maybe I overlooked it).

Points or Answering You (AU): Sustainable Development Goal (SDG) added in full with abbreviation in parenthesis.

Qn: You have many Tables. Will you be putting lines with these tables because they are a little difficult to read without the lines.

AU: All the four tables are formatted to smaller fonts to make the lines readable easy to follow

Qn: Under Discussion, 5th paragraph you mention Paper I for sensitivity analysis. Where would the reader find this if your manuscript was published?

AU: Deleted “Paper I” and replaced it with  “Table 2 above” in the same paragraph

Qn: You mention this in your Limitations and strengths section: that your observational study was nested in a larger study. But are you able to clarify for the reader why you chose 28 weeks? Could you not start earlier?

AU: The reason advanced and “..The inclusion of pregnancies from 28 or more weeks of gestation is based on fetal viability in our low resource settings. Deliveries before 28 weeks of gestational age are considered abortions (in-service personal experience)..”.

Your Complements: Interesting manuscript. I am blessed to live in the USA, so to read that Uganda can do studies, that malaria and HIV can be prevalent, that you actually got respondents to answer was all great.

AU: Thank you so much for your kind complements. Yes in our Uganda, stigma around HIV has greatly reduced due to many USAID research and implementation programs through the years.

Reviewer 2 Report

The authors describe an observational cohort study describing the characteristics of infants born preterm and with low birth weight in post-conflict Uganda. The manuscript is well-written and is of high quality. The study concludes that maternal malaria in pregnancy, history of small newborn and maternal age ≥ 35 increased the likelihood of low birth weight. The incidence of preterm birth was linked to maternal HIV infection, while maternal formal education was associated with a decreased risk of preterm birth. The authors identify some modifiable risk factors that can be addressed to help reduce low birth weight and prematurity. 

- One major comment is about Table 1: the numbers included in the columns do not add up to the column totals for several variables. In addition, for some variables, it appears that the column totals have been flipped between included and excluded groups. Please clarify the numbers and confirm that the statistical analyses included are indeed correct. 

- Minor comments -

1. Line 79: SDG - please expand the abbreviation

2. Line 89: 'followed up to delivery'

3. Line 111: how was history of 'small newborn size' assessed? was this based on a specific weight or an open-ended question?

4. Table 1: Some differences were noted in the included and excluded group based on the numbers provided. Please clarify if these numbers are accurate after revising the table. 

5. Rare minor typographical errors were noticed. recommend please proofreading the manuscript for typos. 

Author Response

Dear Reviewer 2, 

Greetings from Gulu Uganda. Thank you so much for the wonderful comments. I have addressed them all and attach copy hereunder

Thank you so much 

Beatrice 
